# Comprehensive identification of SWI/SNF complex subunits underpins deep eukaryotic ancestry and reveals new plant components

Jorge Hernández-García [1,7], Borja Diego-Martin[1], Peggy Hsuanyu Kuo[2], Yasaman Jami-Alahmadi [3,4], Ajay A. Vashisht[3,4], James Wohlschlegel [3,4], Steven E. Jacobsen [2,5,6], Miguel A. Blázquez[1] & Javier Gallego-Bartolomé [1✉]

Over millions of years, eukaryotes evolved from unicellular to multicellular organisms with increasingly complex genomes and sophisticated gene expression networks. Consequently, chromatin regulators evolved to support this increased complexity. The ATP-dependent chromatin remodelers of the SWI/SNF family are multiprotein complexes that modulate nucleosome positioning and appear under different configurations, which perform distinct functions. While the composition, architecture, and activity of these subclasses are well understood in a limited number of fungal and animal model organisms, the lack of comprehensive information in other eukaryotic organisms precludes the identification of a reliable evolutionary model of SWI/SNF complexes. Here, we performed a systematic analysis using 36 species from animal, fungal, and plant lineages to assess the conservation of known SWI/SNF subunits across eukaryotes. We identified evolutionary relationships that allowed us to propose the composition of a hypothetical ancestral SWI/SNF complex in the last eukaryotic common ancestor. This last common ancestor appears to have undergone several rounds of lineage-specific subunit gains and losses, shaping the current conformation of the known subclasses in animals and fungi. In addition, our results unravel a plant SWI/SNF complex, reminiscent of the animal BAF subclass, which incorporates a set of plant-specific subunits of still unknown function.

[1] Instituto de Biología Molecular y Celular de Plantas (IBMCP), CSIC-Universitat Politècnica de València, Valencia 46022, Spain. [2] Department of Molecular, Cell and Developmental Biology, University of California at Los Angeles, Los Angeles 90095 CA, USA. [3] Department of Biological Chemistry, University of California Los Angeles, Los Angeles 90095 CA, USA. [4] David Geffen School of Medicine, University of California Los Angeles, Los Angeles 90095 CA, USA. [5] Eli & Edythe Broad Center of Regenerative Medicine & Stem Cell Research, University of California at Los Angeles, Los Angeles 90095 CA, USA. [6] Howard Hughes Medical Institute, University of California at Los Angeles, Los Angeles 90095 CA, USA. [7] Present address: Laboratory of Biochemistry, Wageningen University & Research, 6703 WE, Stippeneng 4, Wageningen, The Netherlands. ✉email: jagalbar@ibmcp.upv.es

Eukaryotic life has evolved for millions of years, giving rise to a wide diversity of living forms that range from unicellular organisms to complex multicellular species. The evolution of eukaryotes into complex multicellular organisms has been accompanied by the biological deployment of novel and sophisticated mechanisms to support larger genomes, epigenetic regulation, and intricate gene expression networks. Chromatin plays a prominent role in the regulation of transcriptional states, with the basic chromatin unit being the nucleosome, which consists of four histone pairs (H3, H4, H2A, and H2B) associated with ~147 bp of DNA. Nucleosomes affect many aspects of nuclear biology and act as physical barriers to proteins attempting to access genomic DNA. Moreover, nucleosomes also serve as recruitment platforms for diverse proteins and complexes involved in chromatin regulation. Thus, eukaryotes have evolved numerous ATP-dependent chromatin remodeling complexes to precisely control the nucleosome landscape[1]. These multiprotein complexes contain a catalytic subunit that uses ATP hydrolysis to dissociate DNA from histones, as well as a variety of scaffold and regulatory subunits, that trigger changes in the position or histone composition of nucleosomes. One of the best-understood chromatin remodelers is the SWI/SNF family which was first identified in yeast and is broadly conserved across eukaryotes[2–6]. Malfunction of this remodeler family has a profound impact on important nuclear processes associated with the misregulation of cell differentiation and cancer[2,6,7].

The function, recruitment, and composition of SWI/SNF complexes have been extensively studied in model organisms from yeast to mammals. Detailed characterization of these organisms has revealed that SWI/SNF complexes are predominantly organized in SWI/SNF-BAF and RSC-PBAF subclasses[8,9], as well as the non-canonical BAF (ncBAF) subclass that was recently identified in mammals[10–12]. These subclasses show non-redundant functions and have signature subunits that define each architecture[13,14]. Moreover, as a consequence of gene duplications in multicellular organisms, multiple SWI/SNF subunit paralogs can be selectively incorporated into these complexes in specific cell types or during certain developmental processes, such as in embryonic stem cells and during neuron differentiation[15–17]. This diversity increases the number of possible SWI/SNF complex combinations and provides versatility for specific biological functions. It is thought that the composition and combinatorial assembly of SWI/SNF complexes have evolved in multicellular organisms to accommodate the demands of larger genomes, the presence of new epigenetic regulators like H1 and DNA methylation, and the complex transcriptional regulation required for multicellularity[2].

The composition and architecture of SWI/SNF complexes have been well characterized in *S. cerevisiae* and a handful of animal model organisms. However, equivalent information in other organisms, including plants, remains significantly limited. This gap in information limits our knowledge of how different configurations of SWI/SNF complexes originated and evolved in different taxa. Given the intimate connection between the architecture of the complexes and their functionality, filling this gap is a critical first step in the understanding of SWI/SNF biological functions across eukaryotes.

Here, we leverage the recent increase in genomic and transcriptomic data to describe the evolutionary history of SWI/SNF complexes in eukaryotes. Our comprehensive SWI/SNF subunit search approach across non-model metazoan and fungal species and plants, coupled with in vivo plant-based experiments, has allowed us to (i) establish the degree of conservation of different known subunits, (ii) identify a hypothetical simple complex present in a last eukaryotic common ancestor (LECA) to delineate a possible evolutionary trajectory of different SWI/SNF complexes, and (iii) identify uncharacterized plant-specific SWI/SNF subunits.

## Results

**Identification of SWI/SNF subunits across different eukaryotic lineages.** To investigate the degree of SWI/SNF conservation among eukaryotes, we developed a primary sequence-dependent approach based on a step-by-step phylogeny-driven search of different SWI/SNF subunits. To unify the names of the different subunits across species we used the established HUGO nomenclature (SMARC) and requested the assignment of new HUGO names for those subunits that previously lacked this nomenclature (Table 1). Previously characterized SWI/SNF subunit protein sequences from humans, baker's, and fission yeast were used as starting queries (Table 1 and Supplementary Table 1). We then performed iterative BLASTP/Phmmer searches in the proteomes of phylogenetically-related species. We selected numerous species within Metazoa, Fungi, and Archaeplastida, encompassing a diverse range of species within each lineage phylogeny, with a special focus on the plant kingdom (Supplementary Table 2). We also included two protozoan species with well-annotated genomes: the endo-parasitic kinetoplastid *Trypanosoma brucei*, and the free-living amebozoan *Dictyostelium discoideum* in which no extensive study of SWI/SNF subunits exists to our knowledge. Our analysis identified a set of subunits (SMARCA, SMARCB, SMARCC, SMARCD, and SMARCN) that were unambiguously present in all studied lineages (Fig. 1). We, therefore, conclude that these subunits were part of ancient SWI/SNF complexes of the LECA. Interestingly, we find the protozoan *T. brucei* does not contain orthologs of any of these subunits, suggesting SWI/SNF complexes may be completely absent in this species.

**Plant LFR proteins are true SMARCF orthologs.** Previous studies have proposed a relationship between the metazoan SMARCF1/2 subunits (ARID1 and ARID2) and the fungal Swi1/Rsc9 proteins[18–20]. In animals, SMARCF1 proteins contain a signature amino-terminal AT-rich interaction domain (ARID) domain, followed by an Armadillo fold (ARM). Animal SMARCF2s are characterized by a similar N-terminal structure, including an ARID and ARM domain followed by an RFX DNA-binding domain (DBD), and a C-terminal zinc finger (Fig. 2a). We find that these features are conserved in SMARCF1 and SMARCF2 orthologs in most opisthokonts and that the lack of an automatically detectable ARM domain in *S. cerevisiae* Swi1 and Rsc9 is an exception among fungi (Fig. 2a and Supplementary Fig. 1a). Moreover, ARM and ARID domain phylogenetic analyses support a common evolutionary origin for SMARCF1 and SMARCF2 in animals and fungi (Supplementary Fig. 1b, c), with species-specific motif loss.

Our analyses also suggest *A. thaliana* LFR protein and its plant orthologs share a common ancestry with opisthokonts SMARCF proteins since all of them contain a phylogenetically-related ARM domain. Additionally, we find some chlorophytan algae display a conserved ARID domain in their LFR orthologs (Supplementary Fig. 1a, c). While SMARCF orthologs appear absent in some algal species, this result may be due to lineage-specific losses or poor genome annotation. Additionally, red algae also contain the ARID-ARM architecture in their SMARCF homologs. Therefore, we propose that a SMARCF subunit composed of ARID and ARM domains was also part of an ancestral SWI/SNF complex, and has been maintained in the major lineages of eukaryotes, with limited losses in specific lineages. Lastly, a SMARCF duplication in an opisthokontan common ancestor and the introduction of an RFX DBD domain was associated with the appearance of the

**Table 1 SWI/SNF complex components in reference species.**

| SMARC name | Metazoa (human) | Fungi (baker's yeast/other fungi) | Plants (Arabidopsis) |
|---|---|---|---|
| SMARCA2/4 | BRM, BRG1 | Snf2, Sth1 | BRM, SYD, CHR12, CHR23 |
| SMARCB | BAF47 | Snf5, Sfh1 | BSH |
| SMARCC | BAF155, BAF170 | Swi3, Rsc8 | SWI3A, SWI3B, SWI3C, SWI3D |
| SMARCD | BAF60A, BAF60B, BAF60C | Snf12, Rsc6 | SWP73A, SWP73B |
| SMARCE[a] | BAF57 | Ssr4 (*S. pombe*, Q9P7Y0; *Gonapodya prolifera*, A0A139AVV0) | - |
| SMARCF[b] | ARID1A, ARID1B, ARID2 | Swi1, Rsc9 | LFR |
| *SMARCG* | DPF1, DPF2, DPF3, PHF10 | Swp82, Rsc7(Npl6) | TPF1, TPF2 |
| *SMARCH* | PBRM1 | Rsc1, Rsc2, Rsc4 | - |
| *SMARCI* | BRD7, BRD9 | AGABI2DRAFT_122354 (*A. bisporus*, XP_006456458), UMAB_11035/ UMAG_06029 (*U. maydis*, XP_011392323, XP_011392397), RBRD7/9 (*R. irregularis*, PKK68317) | BRD1, BRD2, BRD13 |
| *SMARCJ* | BCL7A, BCL7B, BCL7C | - | BDH1, BDH2 |
| *SMARCK* | GLTSCR1/1 L | AGABI1DRAFT_112712 (*A. bisporus*, K5W2I3), UMAG_10422 (*U. maydis*, A0A0D1DVN6), RhiirC2_747354 (*R. irregularis*, PKK70026) | BRIP1, BRIP2 |
| *SMARCL* | SS18, CREST, SS18L2 | RiSS18/RirG_105530 (*R. irregularis*, A0A015JMG5) | GIF1, GIF2 |
| *SMARCM* | BCL11A, BCL11B | - | - |
| *SMARCN* | ACTL6A, ACTL6B, ACTB | Arp7, Arp9 | ARP4, ARP7 |
| | - | Snf6 | - |
| | - | Rtt102 | - |
| | - | Rsc3, Rsc30 | - |
| | - | Rsc58 | - |
| | - | Rsc14(Ldb7) | - |
| | - | Htl1 | - |

Non-SMARC names correspond to lineage-specific subunits previously identified.
New SMARC names assigned by the HUGO Gene Nomenclature Committee are indicated in italics.
[a]SMARCE is an animal-specific subunit with already established SMARC nomenclature.
[b]SMARCF naming for ARID-type subunits has been employed in a limited number of databases and articles.
All names correspond to reference species: human for Metazoa, *Saccharomyces cerevisiae* for Fungi, and Arabidopsis for Plants. Subunits not present in the reference species but present in the lineage are indicated.

functionally divergent SWI/SNF-BAF and RSC-PBAF subclasses found in extant fungi and metazoans (Fig. 2b).

**Evolutionary origin of BCL11, PBRM1, BAF57, BRD7/9, BCL7, GLTSCR1, and SS18.** BCL11 was originally identified in multiple metazoan genomes[21]. Despite the expansive searches performed here, we did not identify BCL11 orthologs outside metazoans (Fig. 1). Within Metazoa, BCL11 was absent in the sponge *A. queenslandica* (Porifera), and only chordates presented two BCL11 paralogs (BCL11A/B). These findings suggest BCL11 likely originated in a nephrozoan common ancestor.

The metazoan PBRM1 subunit is characterized by multiple tandem bromodomains and has also been previously associated with the fungal bromodomain Rsc1/2/4 proteins[22,23]. Our analyses confirmed this phylogenetic relationship in multiple lineages, thus we re-named them SMARCH. In contrast, no multiple or single SMARCH bromodomains were found in plants, suggesting this subunit is opisthokontan-specific.

The absence of BAF57-like proteins in *S. cerevisiae* had originally led to the assumption that SMARCE subunits were exclusive to Metazoa[24]. However, other fungal SWI/SNF complexes contain the Ssr4 subunit[25], for which we have found evidence of a shared evolutionary origin with animal BAF57. Although Ssr4 lacks the characteristic HMG domain[24], it contains a region homologous to the animal BAF57 NHRLI domain, which partially overlaps with a previously defined kinesin-like coiled-coil region (KLCC)[24,26] (Supplementary Fig. 2). Importantly, the filasterean *Capsaspora owczarzaki* contains what could be considered an intermediate SMARCE protein that presents characteristics of both metazoan BAF57 and fungal Ssr4, including the N-terminal Ssr4 domain (DUF1750), an HMG domain, and the C-terminal NHRLI domain (Supplementary

Fig. 2). Similar to SMARCH, no homologs for SMARCE were found outside opisthokonts, suggesting SMARCE may have originated in an opisthokontan common ancestor.

We readily identified orthologs of the bromodomain subunit BRD7/9 in most species, with the exceptions of bakers and fission yeasts. In these yeasts, we identified Rsc1/2/4 and Gcn5 among other bromodomain domains as low score hits. In the case of the basidiomycete *Ustilago maydis* and the non-dikaryotic fungus *Rhizophagus irregularis*, we independently identified high confidence SMARCI and SMARCH orthologs, suggesting BRD7/9 was lost within Ascomycota. Similarly, most Archaeplastida species have bromodomain proteins highly similar to BRD7/9. Importantly, the three *A. thaliana* BRDs predicted to be orthologs of HsBRD7/9 (i.e., BRD1, BRD2, and BRD13) have been found to co-immunoprecipitate with other SWI/SNF subunits in vivo and contribute to the proper function of the complex in plants[27–29], supporting orthology with functional conservation. From these findings, we suggest SMARCI also originated in the LECA.

Outside of Metazoa, we find BCL7/SMARCJ orthologs in *D. discoideum* and most plants but not within fungi, suggesting this subunit was lost in an early fungal ancestor (Fig. 1 and Supplementary Fig. 3). The two predicted SMARCJ orthologs in *A. thaliana* that we named BCL-domain homolog 1 (BDH1) and BDH2 (AT4G22320 and AT5G55210, respectively) have been identified with other SWI/SNF subunits in affinity purification experiments[27,30,31]. However, no functional data connecting these proteins to plant SWI/SNF complexes is available. These results suggest the origin of the SMARCJ subunit in the LECA.

GLTSCR1/SMARCK and SS18/SMARCL were also confidently identified in animals, fungi, and plants. SMARCK, a signature subunit of the mammalian non-canonical BAF (ncBAF) complexes, is widespread among nearly all plant species analysed, as

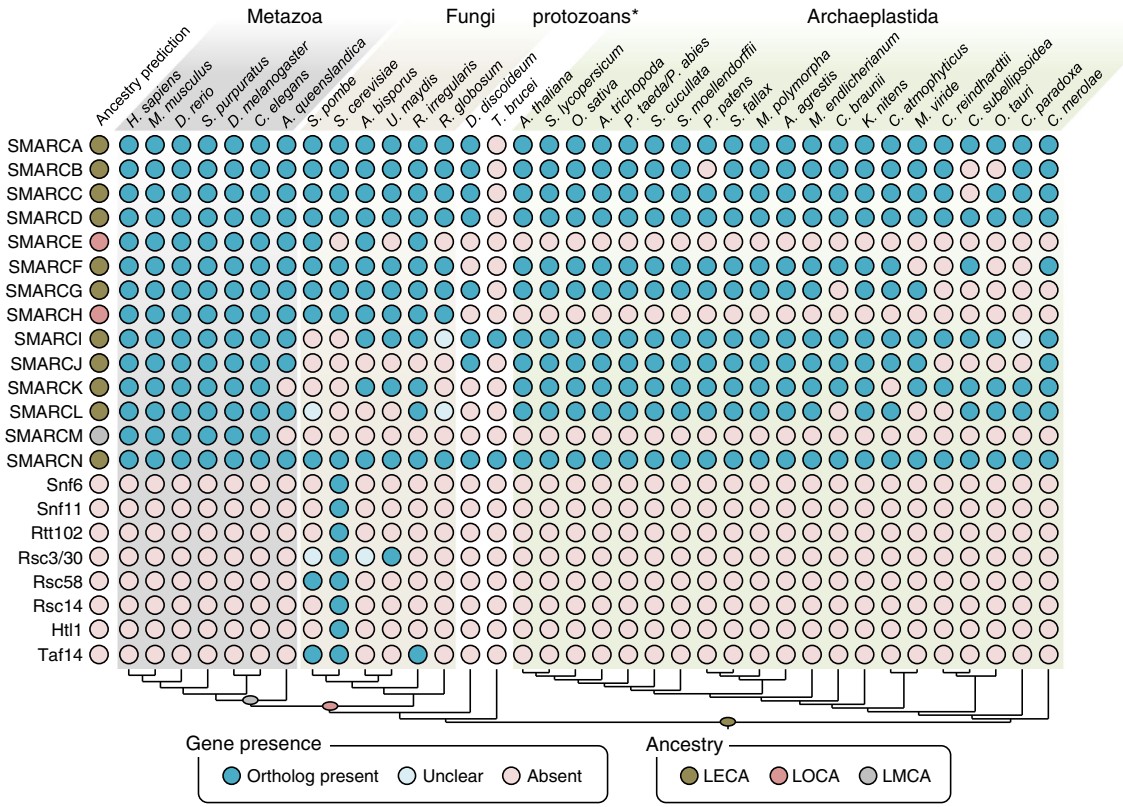

**Fig. 1 Occurrence of SWI/SNF subunits in different eukaryotes.** Blue-filled circles indicate at least a positive hit for the search of a given component (rows) in specific species (columns). Light blue circles indicate the presence of a plausible or distant ortholog hit. Names for each subunit derive from consensus names (see Table 1). The presence of a given homolog was determined as described in Materials and Methods. Ancestral prediction indicates the most likely ancestor that contained a given subunit based on the presence in distinct lineages and species. *protozoan species are entitled together even though they do not form a natural group. The lower cladogram indicates phylogenetic relationships between species. LECA Last Eukaryotic Common Ancestor, LOCA Last Opisthokontan Common Ancestor, LMCA Last Metazoan Common Ancestor, *H. sapiens* Homo sapiens, *M. musculus* Mus musculus, *D. rerio* Danio (Brachydanio) rerio, *S. purpuratus* Strongylocentrotus purpuratus, *D. melanogaster* Drosophila melanogaster, *C. elegans* Caenorhabditis elegans, *A. queenslandica* Amphimedon queenslandica, *S. pombe* Schizosaccharomyces pombe, *S. cerevisiae* Saccharomyces cerevisiae, *A. bisporus* Agaricus bisporus, *U. maydis* Ustilago maydis, *R. irregularis* Rhizophagus irregularis, *R. globosum* Rhizoclosmatium globosum, *D. discoideum* Dictyostelium discoideum, *T. brucei* Trypanosoma brucei, *A. thaliana* Arabidopsis thaliana, *S. lycopersicum* Solanum lycopersicum, *O. sativa* Oryza sativa, *A. trichopoda* Amborella trichopoda, *P taeda* Pinus taeda, *S. cucullata* Salvinia cucullata, *S. moellendorffii* Selaginella moellendorffii, *P. patens* Physcomitrella (Physcomitrium) patens, *S. fallax* Sphagnum fallax, *M. polymorpha* Marchantia polymorpha, *A. agrestis* Anthoceros agrestis, *M. endlicherianum* Mesotaenium endlicherianum, *C. braunii* Chara braunii, *K. nitens* Klebsormidium nitens, *C. atmophyticus* Chlorokybus atmophyticus, *M. viride* Mesostigma viride, *C. reindhardtii* Chlamydomonas reindhardtii, *C. subellipsoidea* Coccomyxa subellipsoidea, *O. tauri* Ostreococcus tauri, *C. paradoxa* Cyanophora paradoxa, *C. merolae* Cyanidioschizon merolae.

well as in fungi, being absent only in ascomycetes (Fig. 1). The two SMARCK subunits we detected in *A. thaliana* have been already characterized as functional components of a SWI/SNF complex that incorporates the *A. thaliana* BRM ATPase (AtSMARCA)[27,30,31]. For SMARCL, we find an SSXT domain-driven phylogenetic connection between SS18 and the previously reported fission yeast-specific Snf30 subunit[25]. The *A. thaliana* SMARCL orthologs we identify belong to the GIF1/AN3 family of transcriptional activators[32] and have previously been associated with plant SWI/SNF complexes[27,30,31]. These results suggest SMARCK and SMARCL were present in the LECA.

**Lineage-specific yeast SWI/SNF subunits.** As a confirmation of the existence of lineage-specific SWI/SNF subunits, we find that ten of the *S. cerevisiae* subunits without reported orthologs in humans or flies indeed had orthologs only in the fungal lineage (Fig. 1). Moreover, a thorough search among all fungal proteomes available in NCBI and MycoCosm suggests many of these subunits can only be found within the order Saccharomycetales—such as Snf6 and Rtt102–, or the family Saccharomycetaceae—such as Lbd7/Rsc14 and Htl1. Rsc3/30-related proteins are

slightly more widespread, as similar sequences are found in ascomycetes and basidiomycetes. However, no experimental evidence supports this potential functional conservation. Taf14 is found in *S. cerevisiae* and *S. pombe* and the distantly related *R. irregularis*, suggesting either a single horizontal transfer event between fungi or multiple loss events during fungal evolution. Rsc58 is confidently found in two species, *S. cerevisiae* and *S. pombe*, and has been functionally associated with the RSC complex in both species[25].

**A common eukaryotic ancestor for the SMARCG subunits.** Based on limited homology searches, the 2x tandem PHD-containing BAF45/SMARCG subunits were considered metazoan SWI/SNF-specific subunits, while the chromatin remodeling complex (CRC)-domain Swp82/Rsc7 subunits were considered fungi-specific[15,33]. Our comprehensive analysis here establishes a previously disregarded common evolutionary origin for both sets of subunits, which is also shared by plants (Fig. 3a and Supplementary Fig. 4). First, the lack of a characteristic PHD domain in the *S. cerevisiae* Swp82 and Rsc7 proteins appears to be an exception among fungi. In fact, most fungal Swp82/Rsc7

# a

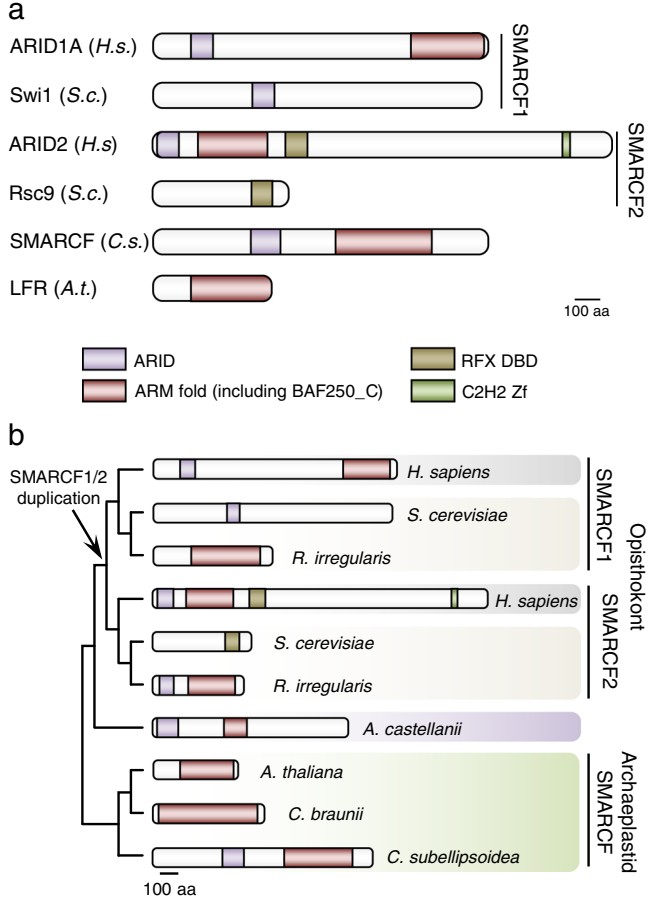

**Fig. 2 SMARCF subunits are found in all eukaryotes. a** Domain architecture of known SMARCF subunits in *H. sapiens*, *S. cerevisiae*, *C. subellipsoidea*, and *A. thaliana*. Scale bar, 100 amino acids. **b** Graphical summary of the evolution of SMARCF domain architectures as predicted from Supplementary Fig. 1. The scale bar indicates the primary sequence length. Phylogram represents the suggested relationship between SMARCF subunits. Arrow indicates the duplication of a single SMARCF into SMARCF1 and SMARCF2 in a fungal and animal common ancestor. Domains are predicted based on Pfam and InterProScan hits, and depicted as colored boxes as indicated in the figure. ARM fold represents a series of ARM-fold hits (IPR016024 and IPR00025) and BAF250_C (PF12031/IPR033388); ARID, AT-rich interaction domain (PF01388/IPR001606); RFX DBD, RFX DNA-binding domain (PF02257/IPR003150); C2H2 Zf, zinc finger, C2H2 type (PF00096). *H.s. Homo sapiens*, *S.c. Saccharomyces cerevisiae*, *R. irregularis Rhizophagus irregularis*, *C.s. Coccomyxa subellipsoidea*, *A. castellanii Acanthamoeba castellanii**, *A.t. Arabidopsis thaliana*, *C. braunii Chara braunii*. **A. castellanii* (Amoebozoa), is a close relative of *D. discoideum* with a bona fide SMARCF subunit.

homologs, such as that of *R. irregularis*, possess highly conserved tandem PHD domains (Fig. 3b). Second, similar PHD-domain proteins are found in plants. These include the proteins AT3G52100 and AT3G08020 in *A. thaliana* that here we have named TRIPLE PHD FINGERS 1 (TPF1) and TPF2, respectively (Fig. 3b, c). Third, the phylogenetic relationship between SMARCG orthologs is likely masked by the presence of distinct domains in different lineages. This suggests a particularly high number of successive chromatin-associated domain gains and losses during the evolutionary history of this protein family (Fig. 3c). For instance, mammalian BAF and PBAF complexes contain two different SMARCG paralogs: PHF10 containing a SAY domain and DPF containing a Req domain[15] (Fig. 3c). In

fungal species such as *R. globosum* and *R. irregularis*, we find proteins that contain not only PHD domains but also Req and CRC domains, reinforcing their identity as SMARCG subunits and their evolutionary relationship to yeast Swp82 and Rsc7. Interestingly, an extended search in all available proteomes suggests SAY, Req, and CRC domains are only found in SMARCG subunits. SAY domains were only present in metazoan PHF10 subunits, while Req domains were also found in several fungal SMARCG proteins (Supplementary Fig. 4b). In turn, we find that CRC domains are fungi-specific and are characteristic of fungal SMARCG (Supplementary Fig. 4c). Additionally, structural predictions of plant SMARCG using Phyre2[34] revealed a C-terminal region that is highly conserved across plants and shows a high similarity to a Tudor histone reader domain[35] (Supplementary Fig. 5a, b). Taken together, our phylogenetic analyses of the PHD, Req, and CRC domains (Supplementary Fig. 4) suggest a complex evolutionary trajectory from an ancestral PHD-containing SMARCG protein with the lineage-specific incorporation of Req (in opisthokonts), CRC (in fungi), SAY (in animals), or Tudor domains (in plants), as well as a hypothesized loss of PHD or Req domains in specific clades.

**SMARCG1/TPF1 is a SWI/SNF subunit in plants**. The plant model organism *A. thaliana* contains two SMARCG subunit paralogs, TPF1 and TPF2. TPF1 displays nuclear sublocalization (Supplementary Fig. 5c) and is able to co-immunoprecipitate multiple known plant SWI/SNF subunits in immunoprecipitation mass spectrometry (IP-MS) experiments (Table 2 and Supplementary Data 1). Of the four plant SWI/SNF ATPases[3], only CHR12 was identified in the TPF1 bait-based IP-MS experiments suggesting TPF1 complexes preferentially incorporate the CHR12 paralog. Interestingly, three highly-enriched uncharacterized TPF1 interactors were previously identified in separate plant IP-MS experiments using different bait SWI/SNF subunits[27,36]. These uncharacterized proteins are SAWADEE HOME-ODOMAIN HOMOLOG 2 (SHH2), and AT1G32730 and AT1G06500, which we refer to as PLANT-SPECIFIC SWI/SNF-ASSOCIATED PROTEIN 1 (PSA1) and PSA2, respectively (Table 2). While no functional data exists for PSA1 and PSA2, SHH2 has been reported to bind H3K9me2 in vitro[37]. Furthermore, when SHH2 is used as an IP-MS bait it pulls down a protein complex similar to the protein complex identified with TPF1 (Table 2 and Supplementary Data 1). Here we identify CHR12 and CHR23 paralogs, as well as the TPF1 paralog TPF2. Moreover, three proteins were highly-enriched: BROMODO-MAIN 5 (BRD5)[38], and two paralogs of a protein we name ONE PHD FINGER 1 (OPF1) and OPF2. Both BRD5 and OPF1 were detected in TPF1 IP-MS experiments. Importantly, these uncharacterized SWI/SNF interactors, SHH2, PSA1, PSA2, BRD5, and OPF1/2, are streptophyte-specific, suggesting lineage-specific subunits have also evolved in plants (Supplementary Fig. 6a). In summary, we have identified a plant SWI/SNF subclass reminiscent of the animal BAF complex that selectively incorporates the ATPases CHR12 or CHR23 as well as a set of plant-specific subunits of still unknown function.

## Discussion

SWI/SNF chromatin remodelers have been studied for decades in different model organisms and a wealth of information is available pertaining to their function, targets, and architecture[2,39]. Our systematic evolutionary approach here confirms the strong conservation of multiple subunits[6,40]. Crucially, our approach here also identifies unnoticed phylogenetic connections between some SWI/SNF subunits and identifies SMARCG as a conserved SWI/SNF subunit in plants. A possible explanation for previously

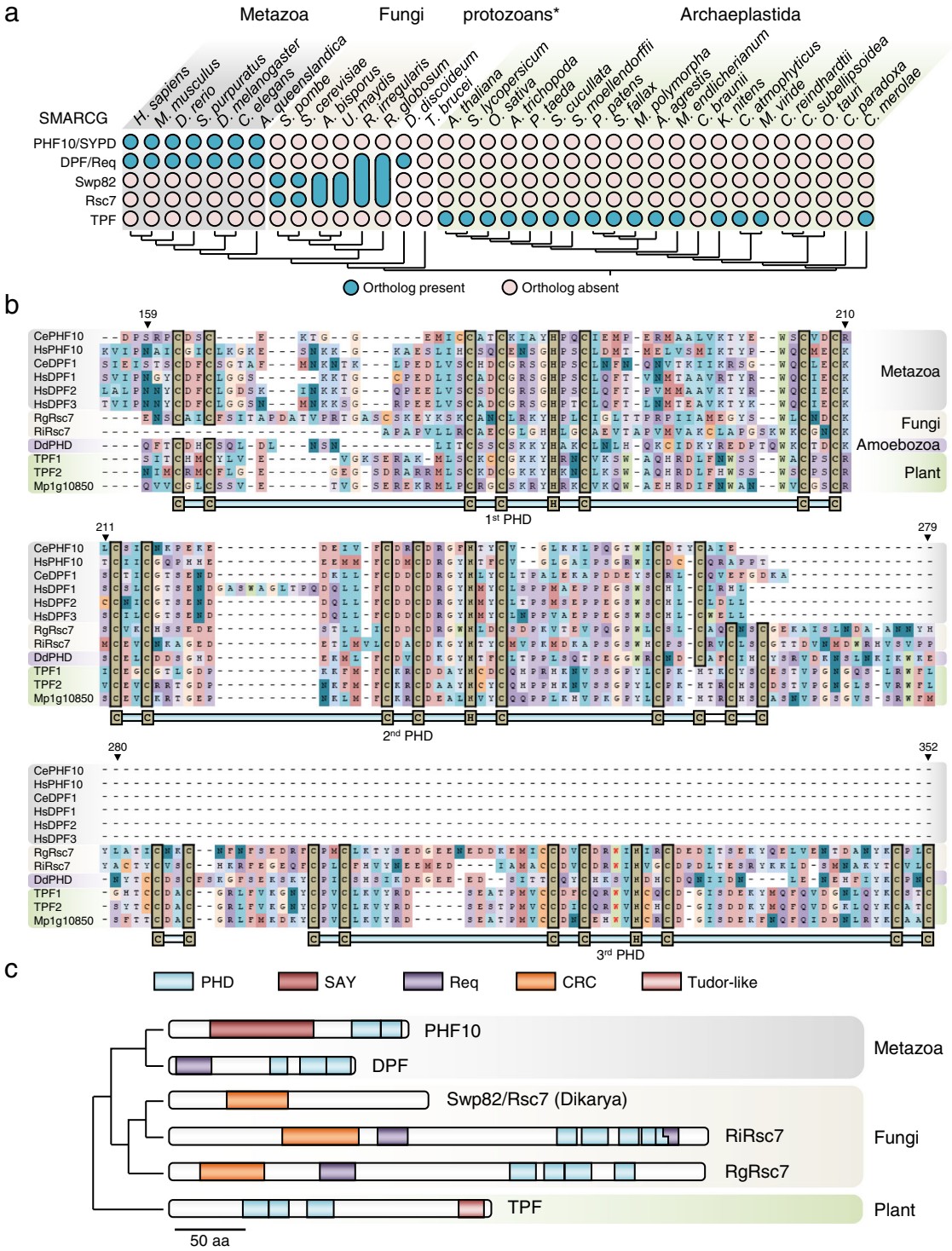

c

| PHD | SAY | Req | CRC | Tudor-like |

missed links is that previous phylogenetic analyses were conducted using a limited number of metazoan, fungal, and plant protein sequences. With this limited dataset, large phylogenetic distances between species could complicate the identification of orthologous genes. Thus, our approach of concatenated searches in the proteomes of phylogenetically-related species is a powerful method to unravel previously hidden phylogenetic relationships across evolutionarily distant species.

Of all previously reported metazoan SWI/SNF subunits, we find only the BAF-specific subunit BCL11[10] lacks homologs in other lineages, while orthologs for all other SWI/SNF subunits

can be found in more distant lineages like fungi and plants. Indeed, our work reveals three phylogenetic relationships that change the evolutionary model of SWI/SNF complexes in eukaryotes: (i) the opistokhontan origin of SMARCE, (ii) the origin of SMARCF in a LECA instead of a last opisthokontan ancestor, and (iii) the origin of SMARCG subunits in a LECA instead of a metazoan ancestor.

Although the BAF57 subunit has been historically defined as an HMG-domain protein specific to Metazoa[24], the presence of an NHRLI domain both in BAF57 and fungal Ssr4 subunits suggests a phylogenetic link and that both proteins have a

**Fig. 3 SMARCG subunits are ancestral PHD-containing proteins. a** Occurrence of SMARCG subunits in different eukaryotes. Blue-filled circles indicate at least a positive hit for the search of a specific SMARCG subtype (rows) in specific species (columns). Each subtype is based on known architectures (PHF10, DPF1-3, and Swp82/Rsc7) or the newly described plant architecture (TPF). Enlarged circles represent subunits with mixed architectures as represented in (**c**) or lack of duplication into Swp82 and Rsc7 subunits. **b** T-Coffee derived multiple sequence alignment of PHD-regions of SMARCG proteins from several eukaryotes. The numbers above columns are referred to residue position in AT3G52100 (TPF1). PHDs structure and relevant residues are indicated below the alignment. Tryptophan residues suggesting binding to methylated H3K4 are marked in red. **c** Summary of the evolution of SMARCG architecture showing the primary structure and domain composition of representative species. Dikaryan SMARCG structure is represented by *Saccharomyces cerevisiae* Swp82 protein due to common domain architecture in Swp82 and Rsc7 proteins. RiRsc7 and RgRsc7, *R. irregularis*, and *R. globosum* SMARCG subunits. Domains are predicted based on Pfam and InterProScan hits, and depicted as colored boxes as indicated in the legend. PHD plant homeodomain (several Pfam hits, clan zf-FYVE-PHD CL0390/IPR019787), SAY supporter of activation of yellow (predicted from bibliography/absent in Pfam/InterProScan), Req requiem/DPF N-terminal domain (PF14051/IPR025750), CRC chromatin remodeling complex Rsc7/Swp82 subunit (PF08624/IPR013933), Tudor-like is derived from several Pfam hits all belonging to the clan Tudor (CL0049), or predicted from multiple sequence alignment. The species list and databases used can be found in Supplementary Table 2. The scale bar indicates the primary sequence length.

### Table 2 Plant BAF subunits found by TPF1 and SHH2 IP-MS.

| Name | Protein ID | TPF1 experiment 1 | | | TPF1 experiment 2 | | | SHH2 experiment 1 | | | SHH2 experiment 2 | | |
|---|---|---|---|---|---|---|---|---|---|---|---|---|---|
| | | Col-0 | #11 | #4 | Col-0 | #11 | #4 | Col-0 | #10 | #21 | Col-0 | #10 | #21 |
| TPF1 | AT3G52100 | - | 36 | 36 | - | 24 | 24 | - | 31 | 30 | - | 31 | 31 |
| ARP7 | AT3G60830 | 24 | 32 | 32 | - | 23 | 21 | 28 | 31 | 31 | 26 | 31 | 31 |
| ARP4 | AT1G18450 | 25 | 30 | 31 | - | 24 | 21 | 27 | 30 | 29 | 25 | 29 | 29 |
| PSA2 | AT1G06500 | - | 30 | 30 | - | 20 | 18 | - | 28 | 28 | - | 29 | 28 |
| SHH2 | AT3G18380 | - | 29 | 29 | - | 20 | 18 | - | 35 | 36 | 24 | 35 | 35 |
| PSA1 | AT1G32730 | - | 29 | 28 | - | 21 | 19 | - | 29 | 28 | - | 29 | 29 |
| BDH1 | AT4G22320 | - | 29 | 30 | - | 20 | 18 | - | 27 | 27 | - | 28 | 27 |
| OPF1 | AT1G50620 | - | 29 | 28 | - | 20 | 18 | - | 33 | 33 | - | 33 | 33 |
| SWI3B | AT2G33610 | - | 29 | 26 | - | 22 | 19 | - | 30 | 29 | - | 30 | 29 |
| SWI3A | AT2G47620 | - | 28 | 26 | - | 23 | 21 | - | 30 | 30 | - | 30 | 30 |
| CHR12 | AT3G06010 | - | 28 | 26 | - | 22 | 20 | - | 30 | 30 | - | 30 | 30 |
| SWP73B | AT5G14170 | - | 28 | 26 | - | 22 | 20 | 25 | 30 | 29 | 25 | 29 | 29 |
| BRD5 | AT1G58025 | - | 27 | 25 | - | 20 | 18 | - | 29 | 27 | - | 29 | 28 |
| LFR | AT3G22990 | - | 25 | 23 | - | 21 | 18 | 23 | 30 | 28 | - | 30 | 30 |
| BSH | AT3G17590 | - | 25 | - | - | 21 | 19 | - | 30 | 29 | - | 30 | 30 |
| OPF2 | AT3G20280 | - | - | - | - | - | - | - | 30 | 30 | - | 30 | 30 |
| TPF2 | AT3G08020 | - | - | - | - | - | - | - | 28 | 27 | - | 29 | 29 |
| BDH2 | AT5G55210 | - | - | - | - | - | - | - | 27 | - | - | 28 | 27 |
| CHR23 | AT5G19310 | - | - | - | - | - | - | - | 27 | 24 | - | 29 | 29 |

Data represents log2 LFQ intensity values of two independent IP-MS experiments in two independent TPF1-3xFLAG and SHH2-3xFLAG lines compared to untransformed Col-0 controls. Shown interactors followed the criteria LFQ control < LFQ transgenic/10 in at least one TPF1-3xFLAG transgenic line in both independent experiments. Corresponding log2 LFQ values for these proteins in the SHH2-3xFLAG experiments are shown on the right. Values for the paralogs OPF2, TPF2, BDH2, and CHR23, which were only detected in the SHH2-3xFLAG experiments, are also included. See Supplementary Data 1 for complete datasets and analyses.

common ancestor that gave rise to the SMARCE family in fungi and animals. In fact, two pieces of evidence support that the NHRLI domain is a reliable indicator of SMARCE conservation. First, two recent cryoEM studies of the mammalian BAF complex were able to resolve only the regions of BAF57 corresponding to the NHRLI domain and the rest of the flanking KLCC domain (amino acids 175–276)[41] or the entire KLCC domain that partially overlaps the NHRLI domain (amino acids 220–298)[42]. These findings suggest a closer relationship between this protein region and other proteins in the complex. Second, the absence of the BAF57 HMG domain does not disrupt SWI/SNF complex chromatin remodeling function in vitro[24], suggesting that other domains have a more prominent functional role. Curiously, it has been suggested that Snf6 may be the functional budding yeast counterpart to metazoan BAF57[43]. However, while we do not find homology between these two proteins, Phyre2-based protein fold prediction analyses of the NHRLI domain in *S. pombe* and *G. prolifera* Ssr4 suggest a similar structure to BAF57 and Snf6 proteins. We speculate the NHRLI domain of Snf6 has significantly diverged from Ssr4 and BAF57, despite all three proteins appearing to perform a similar structural role within the complex. Alternatively, Snf6 and BAF57/Ssr4 could have had different ancestral origins and undergone convergent evolution towards a similar function in the complex.

The common origin of the plant LFR protein and the signature SMARCF subunits Swi1/ARID1 and Rsc9/ARID2 in yeast and animals is supported by our phylogenetic analysis of the ARM domain. Our findings are consistent with the previously described and prominent role the ARM domain plays in the architecture of fungal and animal SWI/SNF complexes[41,44]. Our findings of a single SMARCF protein in plants also confirm the classical SWI/SNF-BAF and RSC-PBAF subclasses of SWI/SNF complexes are likely only present in opisthokonts. Moreover, as plants lack the PBAF/RSC signature SMARCH subunits PBRM1, Rsc1, Rsc2, and Rsc4, we hypothesize PBAF derived from an ancestral "BAF" complex only in the opisthokontan lineage[40] (Fig. 4).

Our findings defining the phylogenetic relationship between metazoan DPF/PHF10, fungal Swp82/Rsc7, and plant TPF proteins highlight the value of accessing multiple transcriptomes in the different phylogenetic lineages. While various fungi have bona fide SMARCG subunits, orthologs in *S. cerevisiae* and the remainder of the Dikarya lineage maintain the characteristic CRC domain but not the tandem PHD fingers, which may have mistakenly led to the conclusion that fungi do not contain SMARCG

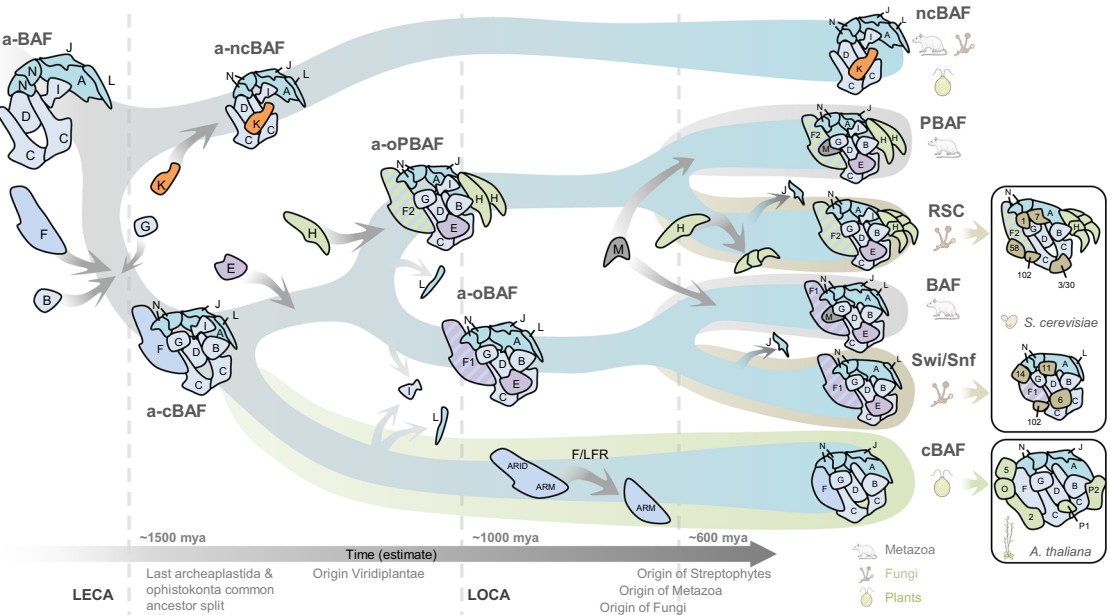

**Fig. 4 Proposed evolution of the SWI/SNF family in eukaryotes.** Model of the evolutionary history of the SWI/SNF complexes from a predicted ancestral complex (a-BAF) composed of an ATPase core module (subunits A, 2xN, L, and J) and a core scaffold module of subunits 2xC, D, and I. Early branching of the a-BAF into an ancestral non-canonical BAF (a-ncBAF) by the acquisition of the K subunit, and an ancestral canonical BAF (a-cBAF) by the addition of B, F, and G subunits most likely occurred in the LECA prior to the split between Archeaplastids and the lineage that gave rise to the opisthokonts. The presence of the K and I signature subunits suggests that ncBAF is present in Metazoa, Archaeplastida, and Fungi (probably lost in derived lineages as Saccharomycetes where K and I are absent). Incorporation of the E subunit into the a-cBAF gave rise to a predicted opisthokontan-type of cBAF that further diverged into the ancestral opisthokontan BAF and PBAF (a-oBAF and a-oPBAF) by the incorporation of the signature H subunit in the PBAF line, and the loss of the L and I in the a-oPBAF and a-oBAF lines, respectively. The duplication and subfunctionalization of the F subunit into the F1 and F2 types of F subunits (for a-oBAF and a-oPBAF, respectively) is also predicted to have occurred during oBAF-oPBAF emergence in the LOCA, before fungi and animals diverged. The Archaeplastid lineage retained a single cBAF that early lost the subunits I and L, and subsequently lost the ARID domain of the F subunit during Streptophyta emergence, originating the LFR type of the F subunit. The acquisition of the M subunit and the loss of the J subunit represent lineage-specific events occurring after the metazoan and fungal split. A putative split of the ancient H subunit into the fungal Rsc1/2/4 subunits could have occurred after the divergence of the a-oPBAF into RSC complexes. The presence of each predicted type of complex in extant lineages is indicated, with lineage-specific subunit composition specified for *S. cerevisiae* and *A. thaliana*, representing SWI/SNF complexes´ flexibility to acquire and lose ancillary subunits. Timeline and indicated events are an estimate. A, SMARCA; B, SMARCB; C, SMARCC; D, SMARCD; E, SMARCE; F, SMARCF (including ARID1, ARID2, and LFR); G, SMARCG (including Swp82/Rsc7, Req/DPFs, and TPFs); H, SMARCH; I, SMARCI; J, SMARCJ; K, SMARCK; L, SMARCL; N, SMARCN (including Actin and Actin-related proteins); 58, Rsc58; 3/30, Rsc3/30; 7, Lbd7; 6, Snf6; 11, Snf11; 102, Rtt102; 1, Htl1; 2, SHH2; P1, PSA1; P2, PSA2; 5, BRD5; O, OPF.

subunits. Interestingly, structural analyses have shown that DPF2 and Rsc7 occupy similar positions in their respective complexes[41,42,45]. On the other hand, the plant SMARCG subunit (TPF) has incorporated a Tudor-like domain, which could be involved in the recognition of methylated histones[35], similar to the described function of PHD domains.

Consistent with its relationship with animal SMARCG sub-units, TPF1 is also found in vivo in a plant SWI/SNF complex that specifically includes the functionally redundant CHR12 or CHR23 ATPases[46], in addition to numerous functionally uncharacterized plant-specific proteins. Among these subunits, functional information only exists for SHH2, which appears to bind H3K9me2 in vitro through its SAWADEE domain[37]. This activity is consistent with its paralog SHH1, which contributes to the plant de novo DNA methylation pathway called RNA-directed DNA methylation (RdDM)[47]. While SHH1 appears to be angiosperm-specific, SHH2 orthologs are found in most strep-tophytes (Supplementary Fig. 6b). The failure to identify any RdDM components by SHH2 IP-MS suggests SHH2 has ancestral functions in the SWI/SNF complex and implies a neofunctiona-lization of SHH1 in RdDM. Furthermore, our IP-MS findings that TPF1 bait fails to identify TPF2, while SHH2 bait identified both TPF1 and TPF2, suggests that this SWI/SNF complex only utilizes

one TPF protein. We extend this finding to OPF paralogs. Future functional studies will help clarify these questions.

Importantly, none of the uncharacterized putative SWI/SNF subunits we identify here (SHH2, PSA1, PSA2, BRD5, and OPF1/2), nor TPF1/2 have previously been identified in published experiments using the plant BRAHMA (BRM) ATPase as bait, or even other proteins that interact with BRM, such as SWI3C, BRIP1/2, BRD1/2/13, and SS18/GIF1[27,28,31] (Table 1). However, recent BRM bait experiments detected plant SMARCK (BRIP1/2), SMARCI (BRD1/2/13), and SMARCL (GIF2) that were not found in TPF1 or SHH2 bait-based IP-MS experiments[28,31]. These results suggest plants have at least two different SWI/SNF sub-classes (Fig. 4). One subclass would specifically incorporate the PHD-containing proteins TPF1/2 and OPF1/2, the bromodomain protein BRD5, SHH2, PSA1 and PSA2, the CHR12 or CHR23 ATPases, and SMARCF/LFR. A second subclass would incorpo-rate the BRM ATPase, together with the BRIP1/2, BRD1/2/13, and SS18/AN3 subunits (Fig. 4). It's worth noting that the composition of these subclasses is reminiscent of metazoan BAF-PBAF and ncBAF, respectively (Supplementary Fig. 6c). Consistent with these findings, SMARCF/LFR, the BAF/PBAF signature subunit, is detected in TPF1 and SHH2 bait experi-ments but is not identified in two recent studies using BRM as

bait[28,31]. In summary, we have identified a plant canonical BAF-like (cBAF) complex that likely evolved from the ancestral form that branched into the SWI/SNF-BAF and RSC-PBAF complexes during opisthokontan evolution, and subsequently incorporated a set of plant-specific subunits of yet unknown function.

The broad conservation of multiple subunits across diverse lineages suggests that SWI/SNF complexes have been conserved through millions of years as important factors for controlling chromatin accessibility. During the evolution of specific lineages, these remodeler complexes have diverged into different subclasses and have incorporated species-specific subunits, resulting in a diverse array of architectures and compositions in the extant lineages. A notable exception appears to be the loss of subunits in the parasite *T. brucei*. However, this might be explained by the preference in this organism for posttranscriptional regulation of polycistronic genes as a mechanism of regulating gene expression[48]. How SWI/SNF complexes evolved into their current architectures and what was the composition of the ancient SWI/SNF in LECAs remains to be described in detail. Our evolutionary analyses across multiple lineages allowed us to speculate about the composition of the SWI/SNF complex in the LECA. From these data, we propose a model for the evolution of the SWI/SNF complexes (Fig. 4) where an ancient complex in LECA first diverged into two main subclasses (a-cBAF and a-ncBAF), present in all extant eukaryotes, followed by divergence of a-cBAF into two separate subclasses (a-oBAF and a-oPBAF) characteristic of only the opisthokontan lineage. Subsequently, we suggest further loss and incorporation of lineage-specific subunits shaped the current architectures of extant SWI/SNF complexes. We hope our evolutionarily informed model will facilitate future SWI/SNF functional analyses across a broad range of species.

## Methods

### Identification of SWI/SNF complex subunit sequences in eukaryotes.
Ortholog searches for each subunit in the SWI/SNF complex were performed following a phylogeny-based step-by-step look-up. Previously characterized protein sequences from the subunits of the *Homo sapiens* BAF-PBAF-ncBAF complexes, and its orthologous complexes in baker's yeast (Swi/Snf2 and RSC) were used as starting queries (Table 1). All the resulting hits were repeatedly used as queries in new searches until no new hits appeared. A combination of BLASTP and Phmmer searches using protein sequences as queries were performed on proteome databases for a range of selected species within the eukaryotic phylogeny, giving particular emphasis to the green plant lineage (Supplementary Table 2). Instead of establishing a priori BlastP score thresholds for all proteins, we relied on the following procedure: First, we performed reciprocal BLAST using the subjects as queries and kept the hits that appeared in both directions (also known as the best bi-directional BLAST hit strategy). Then we performed Pfam (http://pfam.sanger.ac.uk/search) and InterProScan (https://www.ebi.ac.uk/interpro) analyses to confirm the presence of known domains and protein architectures. HMM searches using Phmmer were also performed for several subunits. These searches served both as complementary confirmation, and to identify distantly related orthologs. Robust hits that did not match the expected domain composition were used to propose previously overlooked connections between sequences, attributable to domain loss/gains. When no hits were found, taxa sampling was extended to entire lineage proteomes in NCBI, UniProt, Phytozome, Mycocosm, and oneKP databases to confirm the absence or presence of specific proteins or domains. Manual curation was performed for inconclusive or unclear cases. Finally, positive hits were included in the BLASTP/Phmmer query list when these were successfully found, restarting the search process.

### Phylogenetic analysis.
For SMARCF, ARID domain sequences were extracted based on Pfam/InterProScan automatic annotation (ARID, PF01388/IPR001606). ARID sequences shorter than 50 amino acids were curated with flanking sequences by MAFFT alignment with close orthologs if possible or otherwise discarded. Related ARID domains from JARID, ARID4, ARID3, and other proteins were included as outgroups. ARM-fold sequences were delimited using ARM-type fold derived sequences as a scaffold (IPR016024), or ARM-like (IPR011989) in the case of *Danio rerio* ARID1A, and curated with flanking sequences aided by both BAF250_C (PF12031) and Arm (PF00514) annotations. Unannotated ARMs were predicted by MAFFT alignment to close orthologs when possible. Additional ARM-folds from several species were included as outgroups. For SMARCG, PHD domain sequences were extracted based on multiple Pfam/InterProScan annotations (IPR001965; Clan zf-FYVE-PHD CL0390/IPR019787, including PF00096,

PF00130, PF00628, PF13639, PF13831, PF13894, PF14446, PF15446, PF16866, and PF18112) and were merged to remove duplicates and small hits (<50 amino acids). Similarly, Req (Requiem/DPF N-terminal domain, PF14051/IPR025750), and CRC (chromatin remodeling complex Rsc7/Swp82 subunit, PF08624/IPR013933) sequences were extracted, curated, and predicted from multiple sequence alignments in unannotated sequences in the case of divergent fungal Req sequences. Final alignments were performed with M-Coffee using a combination of the multiple alignment methods MAFFT, T-Coffee, MUSCLE, and POA2[49] followed by manual curation. Trimming was performed around unambiguously aligned regions. Best-fit models of amino acid replacement were selected using the AIC model for ranking, being LG model for CRC (LG + F), Req, and ARM-fold domain trees, WAG model for the ARID domain tree, and JTT model for PHD-domain tree. These models were used to construct unrooted maximum-likelihood trees with PhyML v3.1[50], using empirically estimated amino acid frequencies when indicated (+F). Maximum-likelihood bootstrap support was calculated with 1000 replicates. The graphical representation of the phylogenetic trees was generated using FigTree (version 1.4.3) software (http://tree.bio.ed.ac.uk/software/figtree/), and the final figures were edited manually.

### Plant materials.
A genomic fragment of TPF1 including 1137 bp of the promoter and up to the codon before the stop codon was amplified using the oligos fwd 5′-CACCCGAAGAGAGAAACTTATGTACCTC-3′ and rev 5′-CGTTCTCGTTCTCTTTTGTTTC-3′ and cloned in a pENTR/D plasmid (Invitrogen) to create pENTR-TPF1. Similarly, a genomic fragment of SHH2 including 1294 bp of promoter up to the codon before the stop codon was amplified using the oligos fwd 5′-CACCCCGTCTTCATGTCACGAGCAAGC-3′ and 5′-GGCTGAACCAGCGG-GAACAGTAGC-3′ and cloned in a pENTR/D plasmid (Invitrogen) to create pENTR-SHH2. The TPF1 genomic fragment was transferred by an LR reaction (Invitrogen) to a modified pEG302 plasmid that introduces a C-terminal 3xFLAG tag to generate pEG302-TPF1-3xFLAG. A 3xFLAG tag was introduced in the AscI site of pENTR-SHH2 to create pENTR-SHH2-3xFLAG. The resulting plasmid was introduced in a modified pEG302 by an LR reaction to generate pEG302-SHH2-3xFLAG. The pEG302-TPF1-3xFLAG and pEG302-SHH2-3xFLAG plasmids were transformed into the *tpf1-1* T-DNA insertion line (SALK_010411C) and Col-0 wild-type plants, respectively, by the floral dip method.

### IP-MS.
Immunoprecipitations were performed as previously described[51] and were identical for the TPF1-3xFLAG and SHH2-3xFLAG experiments. Two independent transgenic lines expressing TPF1-3xFLAG or SHH2-3xFLAG, as well as a Col-0 control, were grown in a greenhouse under long-day conditions. Eight grams of inflorescences were collected, ground in liquid nitrogen, and resuspended in 40 ml of IP buffer (50 mM Tris pH 7.6, 150 mM NaCl, 5 mM MgCl₂, 10% glycerol, 0.1% NP40, 0.5 mM DTT, 1 mM PMSF, 1 µg/µL pepstatin, and 1× Complete EDTA-Free (Sigma). After a filter with one layer of Miracloth (Merck, cat#475855), samples were homogenized with a douncer (ten times soft and ten times hard), centrifuged at 4 °C for 10 min at 10,000 × g, and filtered using a 40 µm cell strainer. Samples were rotated for 3 h at 4 °C with 200 µl of Anti-FLAG M2 magnetic beads (Sigma, cat#M8823) that were previously blocked with 5% BSA. The magnetic beads were captured and washed four times with IP buffer and two times with IP buffer without NP40. Samples were eluted for 30 min with 150 µl of 3xFLAG peptide (Sigma, cat#F4799) at a concentration of 250 µg/ml in IP buffer without NP40. This was repeated two more times and the eluates were combined. Proteins were precipitated by the addition of TCA to a final concentration of 20% and incubated for 30 min on ice followed by a 30 min 4 °C centrifugation at 12,000 × g. Samples were washed three times with 250 µl of cold acetone and after the last wash, the pellet was air-dried.

### LC-MS methodology.
For experiment 1 of TPF1-3xFLAG and the SHH2-3xFLAG experiments, the samples were prepared and analysed at the Wohlschlegel Lab at UCLA (Los Angeles, US). Briefly, proteins were reduced and alkylated using 5 mM Tris (2-carboxyethyl) phosphine and 10 mM iodoacetamide, respectively. Protein digestion was achieved by sequential addition of endopeptidase Lys-C (BioLabs) and Trypsin (Pierce™) at a 1:100 enzyme/protein ratio and incubated at 37 °C overnight. The digested samples were quenched by the addition of formic acid to the 5% (v./v.) final concentration. Finally, desalting prior to LC-MS/MS analysis was done using C18 pipette tips (Thermo Scientific, cat# 87784) and reconstituted in 5% formic acid before being analysed by LC-MS/MS. Peptide mixtures were fractionated online using a 25 cm long, 75 µm ID fused-silica capillary that was packed in-house with bulk ReproSil-Pur 120 C18-AQ particles as described elsewhere[52]. Peptides were subjected to a 140-min water-acetonitrile linear gradient in 6–28% buffer B (acetonitrile solution with 3% DMSO and 0.1% formic acid) at a flow rate of 200 nl min⁻¹ which was further increased to 35% followed by a rapid ramp-up to 85% using a Dionex Ultimate 3000 UHPLC (Thermo Fisher Scientific). The eluted peptides were then ionized via nanoelectrospray ionization, and mass spectrometry data were acquired using an Orbitrap Fusion™ Lumos™ Tribrid™ Mass Spectrometer (Thermo Fisher Scientific) with an MS1 resolution of 120,000 followed by sequential MS2 scans at a resolution of 15,000. For the SHH2-3xFLAG samples, trypsin-digested protein samples were analyzed by tandem mass spectrometry using a Thermo Easy-nLC system coupled to a Thermo Q-Exactive MS. For experiment 2 of TPF1-3xFLAG, samples were prepared and analysed at the

Proteomics Facility at the Research Support Central Service at the University of Cordoba (Cordoba, Spain) as previously described[53]. Briefly, protein extracts were cleaned-up in 1D SDS-PAGE at 10% of polyacrylamide. Protein bands were firstly distained in 200 mM ammonium bicarbonate (AB)/50% acetonitrile for 15 and 5 min in 100% acetonitrile. Proteins were reduced by the addition of 20 mM dithiothreitol in 25 mM AB and incubated for 20 min at 55 °C, followed by alkylation of free thiols through the addition of 40 mM iodoacetamide in 25 mM AB in the dark for 20 min. After, protein bands were washed twice in 25 mM AB. Proteolytic digestion was performed by addition of Trypsin (Promega, Madison, WI), 12.5 ng/ul of enzyme in 25 mM AB, and incubated at 37 °C overnight. Protein digestion was stopped by the addition of trifluoroacetic acid at 1% final concentration. Digest samples were dried in a speedvac. Nano LC was performed in Dionex Ultimate 3000 nano UPLC (Thermo Scientific) with a C18 75 µm × 50 Acclaim Pepmam column (Thermo Scientific). Previously, peptide mixes were loaded in a 300 um × 5 mm Acclaim Pepmap precolumn (Thermo Scientific) in 2% acetonitrile/0.05% TFA for 5 min at 5 ul/min. Peptide separation was performed at 40 °C for all runs. Mobile phase buffer A was composed of water, 0.1% formic acid. Mobile phase B was composed of 20% acetonitrile, 0.1% formic acid. Samples were separated at 300 nl/min. Mobile phase B increases to 4–45% B for 60 min; 45–90% B for 1 min, followed by a 5 min wash at 90% B and a 15 min re-equilibration at 4% B. The total time of chromatography was 85 min. The eluted peptides were then ionized via nanoelectrospray ionization, and mass spectrometry data were acquired using an Orbitrap Fusion™ Tribrid™ Mass Spectrometer (Thermo Fisher Scientific) with an MS1 resolution of 120,000 followed by sequential MS2 scans in the ion trap with CID fragmentation and rapid scan mode.

For all LC-MS experiments, raw data were searched against the TAIR Arabidopsis reference proteome. Label-free quantitation (LFQ) intensities were calculated by applying the default settings for LFQ analysis using MaxQuant software as described previously[54].

**Transient expression and confocal microscopy**. The pENTR-TPF1 plasmid described above was transferred by LR reaction (Invitrogen) to a pMDC107[55] to create pMDC107-TPF1. The pMDC107-TPF1 and pBin61-P19 plasmids were transformed into *Agrobacterium tumefaciens* strain *GV3101 C58C1*. Transformants were grown on a selective medium and resuspended in agroinfiltration solution (10 mM MES pH 5.6, 20 mM MgCl$_2$, 200 µM acetosyringone) and incubated for 2 h at room temperature. Bacterial suspensions were then mixed adjusting to a final OD600 of 0.1 and 0.01 for pMDC107-TPF1 and pBin61-P19 cultures, respectively. Forty-eight hours post-infiltration, leave disks were collected, incubated for 10 min with 5 µg/ml DAPI, and imaged using an Axio Observer 780 Confocal microscopy (Zeiss).

**Structure prediction analysis of TFP1**. Phyre2[34] was used to predict the structure of the TPF1 (AT3G52100) C-terminal domain using amino acids 631 to 688 as a template and job type = intensive. The AlphaFold predicted structure corresponding to the same region was downloaded from the AlphaFold Protein Structure Database (F4J5R1)[56]. Human 53BP1 structure was downloaded from PDB (ID c1xniI), and residues corresponding to the Tudor domain (1485-1537) were extracted. Molecular graphics and analyses were performed using ChimeraX[57]. Superimposed structures and root-mean-square deviations (RMSD) values were obtained using the built-in align command.

**Statistics and reproducibility**. Pfam and InterProScan default cut-offs were used for domain inference unless specified. Statistical significances supporting branches in phylogenetic trees were performed by maximum-likelihood bootstrap analysis of 1000 replicates in each tree. For IP-MS experiments, two independent experiments were done where two independent transgenic lines and one control line were used.

**Reporting summary**. Further information on research design is available in the Nature Research Reporting Summary linked to this article.

## Data availability
The datasets generated during and/or analyzed during the current study are available in Mendeley at https://data.mendeley.com/datasets/6m4b8zrnpt/ and in the Supplementary Data 1 file.

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

## Acknowledgements
We thank Ceejay Lee and Nathan Cai for their technical support. We thank Drs Jake Harris and Bruno Catarino for their critical reading of the manuscript. This work was supported by grants NIH Grant R35 GM130272 [to S.E.J.], RYC2018-024108-I [to J.G.-B.] funded by MCIN/AEI/10.13039/501100011033 and by "ESF Investing in your future", and PID2019-108577GA-I00 [to J.G.-B.] and PID2019-110717GB [to M.A.B.] funded by MCIN/AEI/10.13039/501100011033. S.E.J. is a Howard Hughes Medical Institute investigator.

## Author contributions
J.H.-G. conducted all phylogenetic analyses. B.D.-M. and J.G.-B. conducted the confocal microscopy experiment. P.H.-K. and J.G.-B. generated transgenic lines. B.D.-M., P.H.-K., and J.G.-B. conducted the immunoprecipitation experiments. Y.J.-A., A.A.V., and J.W. conducted the mass spectrometry analyses. S.E.J supervised the research and edited the manuscript. J.H-G., M.A.B., and J.G.-B. conceived the original experimental design and wrote the paper.

## Competing interests
The authors declare no competing interests.
