## [Peer Review File · Communications Biology]

Reviewers' comments:

Reviewer #1 (Remarks to the Author):

The manuscript by Hernández-García et al. provides a comprehensive identification and analysis of SWI/SNF complex subunits across eukaryotes. Authors proposed a hypothetical ancestral SWI/SNF complex in the last eukaryotic common ancestor by identifying novel evolutionary relationships, and unraveled a novel plant SWI/SNF complex with unknown function. The finding might facilitate future SWI/SNF functional analyses across a broad range of species. However, there are some items need be addressed in order to strengthen their conclusion before its consideration for publication.

1. In the "Material and Methods" part, line 382 to line 399, authors described the orthologous identification methods. However, they did not provide the details of orthologous criterion, like sequence alignment ratio, BlastP score... Besides, authors only use baker's yeast (Swi/snf2 and RSC) as starting queries is not enough. Because the sequence varies greatly among different lineages, only one starting might lead to incomplete identification. Authors should use more starting queries, for example, select a starting query for each lineage.

2. Authors identified a set of plant-specific subunits with unknown function. Could authors provide some genetic evidence for their function, even in model plant Arabidopsis that is easy to perform functional analysis?

Reviewer #2 (Remarks to the Author):

The Authors reassess the phylogeny of SWI/SNF (also called BAF) chromatin remodeling complexes across a wide range of eukaryotes.

The study clarify the ancestry and the fact that after an early divergence most group retain two types of complexes, an ancestral type and a divergent, group-specific complex.

The analysis uncovers as well a few additional subunits specific to plants. Overall, the work will be useful for researchers in the field.

Minor modifications recommended:

1. The nomenclature of the complexes is confusing and the only thing I would urge the Authors to do is to avoid more confusion.

The new names proposed UNK1/2 and OPF1 certainly do not help in this respect and it would be great to find alternatives that recall directly the origins of these proteins.

2. Currently the manuscript reads like a list of facts. The manuscript would profit from clarifying clearly in the introduction what are the major gaps in knowledge in the field and attempt to write with a little more function in mind.

We would like to thank the two reviewers for their positive comments and suggestions. Please find our replies in blue below each comment.

R1:

The manuscript by Hernández-García et al. provides a comprehensive identification and analysis of SWI/SNF complex subunits across eukaryotes. Authors proposed a hypothetical ancestral SWI/SNF complex in the last eukaryotic common ancestor by identifying novel evolutionary relationships, and unraveled a novel plant SWI/SNF complex with unknown function. The finding might facilitate future SWI/SNF functional analyses across a broad range of species. However, there are some items need be addressed in order to strengthen their conclusion before its consideration for publication.

1. In the “Material and Methods” part, line 382 to line 399, authors described the orthologous identification methods. However, they did not provide the details of orthologous criterion, like sequence alignment ratio, BlastP score... Besides, authors only use baker’s yeast (*Swi/snf2* and *RSC*) as starting queries is not enough. Because the sequence varies greatly among different lineages, only one starting might lead to incomplete identification. Authors should use more starting queries, for example, select a starting query for each lineage.

We apologize for not being totally clear with the description of the procedure, but we had indeed followed the strategy suggested by the reviewer. We have changed the text and include the details:

- The initial search was done using bona-fide SWI/SNF subunits from three species (ie, *H. sapiens*, *S. cerevisiae* and *S. pombe*), where they had been unambiguously described. And then we continuously introduced every confirmed hit as a new query in the search. As indicated by the reviewer, this is the most suitable approach to identify distant orthologs of known subunits in all lineages; but not subunits with absolutely no resemblance to known proteins.
- We have now explained more explicitly in the text the criteria used to select orthologs during the search. On one hand, the thresholds for BlastP scores cannot be fixed beforehand (they depend on the protein family under study); and we do not use alignment scores because they are not a faithful criterion for orthology. What we do is: (1) reciprocal Blasts (between the query and the putative hits), and we keep those hits that appear in both directions; (2) Pfam and InterPro searches with the selected hits, and we keep those that yield the expected domain assignment; (3) for those cases in which we find significant reciprocal Blast hits, but Pfam and InterPro searches do not identify previously known motifs, we inspect the alignments and this leads us to establish previously overlooked connections between proteins due, for instance, to alteration in domain structure. These cases are explained in the text.
- The best guarantee that the selected hits are orthologs is the branch support obtained from the statistical analysis in the phylogenetic trees.

2. Authors identified a set of plant-specific subunits with unknown function. Could authors provide some genetic evidence for their function, even in model plant *Arabidopsis* that is easy to perform functional analysis?

Thanks for the suggestion. *Arabidopsis* is fairly amenable to molecular genetics but the kind of evidence required to assign new functions is not trivial, and time consuming. We are currently performing multiple genetics, molecular biology, and genomics experiments to try to understand the molecular function of these subunits within the complex. These are the main lines in our lab and we hope to publish these results in the future. However, publishing here a fraction of our results would certainly damage the novelty of our future manuscripts while not adding much to the current manuscript whose main scope is the study of the conservation and evolution of SWI/SNF complexes across eukaryotes and not the functional characterization of specific subunits in one species.

R3:

The Authors reassess the phylogeny of SWI/SNF (also called BAF) chromatin remodeling complexes across a wide range of eukaryotes.

The study clarifies the ancestry and the fact that after an early divergence most groups retain two types of complexes, an ancestral type and a divergent, group-specific complex.

The analysis uncovers as well a few additional subunits specific to plants. Overall, the work will be useful for researchers in the field.

Minor modifications recommended:

1. The nomenclature of the complexes is confusing and the only thing I would urge the Authors to do is to avoid more confusion.

The new names proposed UNK1/2 and OPF1 certainly do not help in this respect and it would be great to find alternatives that recall directly the origins of these proteins.

Thanks for this suggestion. We completely agree that the nomenclature of the SWI/SNF subunits is confusing and not very friendly. The HUGO gene nomenclature committee (<https://www.genenames.org>) has previously named a number of SWI/SNF subunits with the *SWI/SNF-related, matrix-associated, actin-dependent regulators of chromatin* (SMARC) nomenclature (ie SMARCA, SMARCB, etc...). However, not all human SWI/SNF subunits were named with this nomenclature. In order to promote a universal nomenclature for SWI/SNF subunits, we contacted the HUGO committee to request new SMARC names for the rest of SWI/SNF subunits that were not previously named and that were conserved across lineages (SMARCG to SMARCN, Table 1). The committee agreed and these new names are already available in the HUGO database. We hope that the community starts using this nomenclature to refer to SWI/SNF subunits.

In this manuscript we named 5 uncharacterized SWI/SNF plant subunits (TPF, OPF, BDH, UNK1 and UNK2). Out of these, only TPF (SMARC name: SMARCG) and BDH (SMARC name: SMARCJ) present orthologs in animal and/or fungal lineages (Table 1). The animal orthologs of TPF are named DOUBLE PHD FINGERS (DPF) based on their characteristic domain composition (2xPHD). To be consistent with this nomenclature, we named this new subunit TRIPLE PHD FINGERS (TPF) after its 3xPHD composition. In the case of the BCL-domain homolog (BDH) subunit, we named it after the only recognizable domain –BCL- previously described in their animal counterparts.

For OPF and UNK1/2 there are no orthologs in other lineages beyond the plant lineage. Consistent with our criteria to name TPF and BDH proteins after their characteristic domains, we decided to name these new subunits after their known domains. In the case of ONE PHD FINGER (OPF), we wanted to be consistent with the nomenclature of other PHD-containing proteins in the SWI/SNF complex (TPF and DPF). Thus, we decided to name it after its characteristic domain (1xPHD). In the case of UNKNOWN1 (UNK1) and UNKNOWN2 (UNK2) there are no identifiable domains or other features to refer to. Thus, we decided to name them UNKNOWN proteins. To follow the reviewer's advice, we decided to rename UNK1 and UNK2 based on their origin. In this case we propose the name PLANT-SPECIFIC SWI/SNF-ASSOCIATED PROTEIN (PSA). Thus, we replaced UNK1 for PSA1, and UNK2 for PSA2 in both text and figures.

2. Currently the manuscript reads like a list of facts. The manuscript would profit from clarifying clearly in the introduction what are the major gaps in knowledge in the field and attempt to write with a little more function in mind.

We thank the reviewer for this comment. In order to clarify the major gaps of knowledge and justify our approach, we modified lines 95-100 in the introduction:

"However, equivalent information in other organisms, including plants, remains significantly limited. This gap of information limits our knowledge of how different configurations of SWI/SNF complexes originated and evolved in different taxa. Given the intimate connection between the architecture of the complexes and their functionality, filling this gap is a critical first step in the understanding of SWI/SNF biological functions across eukaryotes."

We share with the reviewer the urge to acquire more functional information on SWI/SNF complexes in lineages not previously covered by model species. However, we consider that it is rather risky to connect phylogenetic information with functional implications. For instance, absence of one particular subunit in a given organism would not imply that the function of that subunit is missing in this organism, since it might be performed by another -identified or yet-unidentified- subunit. This is why here we strictly present the data that refer to evolutionary relationships between SWI/SNF subunits in the different taxa examined, to set a solid framework for future experimental studies by the community -which are the only vehicle to acquire functional information.

REVIEWERS' COMMENTS:

Reviewer #1 (Remarks to the Author):

The main text and figures had been substantially improved in the revised version, although they are arguing the functional analysis of candidate genes from last review.